# Quantum Spin Hall Effect in Two-Monolayer-Thick InN/InGaN Coupled Multiple Quantum Wells

**DOI:** 10.3390/nano13152212

**Published:** 2023-07-30

**Authors:** Sławomir P. Łepkowski

**Affiliations:** Institute of High Pressure Physics—Unipress, Polish Academy of Sciences, ul. Sokołowska 29/37, 01-142 Warszawa, Poland; slawomir.lepkowski@unipress.waw.pl; Tel.: +48-510-332-238

**Keywords:** quantum spin Hall effect, topological insulators, group-III nitrides, coupled quantum wells

## Abstract

In this study, we present a theoretical study of the quantum spin Hall effect in InN/InGaN coupled multiple quantum wells with the individual well widths equal to two atomic monolayers. We consider triple and quadruple quantum wells in which the In content in the interwell barriers is greater than or equal to the In content in the external barriers. To calculate the electronic subbands in these nanostructures, we use the eight-band **k∙p** Hamiltonian, assuming that the effective spin–orbit interaction in InN is negative, which represents the worst-case scenario for achieving a two-dimensional topological insulator. For triple quantum wells, we find that when the In contents of the external and interwell barriers are the same and the widths of the internal barriers are equal to two monolayers, a topological insulator with a bulk energy gap of 0.25 meV can appear. Increasing the In content in the interwell barriers leads to a significant increase in the bulk energy gap of the topological insulator, reaching about 0.8 meV. In these structures, the topological insulator can be achieved when the In content in the external barriers is about 0.64, causing relatively low strain in quantum wells and making the epitaxial growth of these structures within the range of current technology. Using the effective 2D Hamiltonian, we study the edge states in strip structures containing topological triple quantum wells. We demonstrate that the opening of the gap in the spectrum of the edge states caused by decreasing the width of the strip has an oscillatory character regardless of whether the pseudospin-mixing elements of the effective Hamiltonian are omitted or taken into account. The strength of the finite size effect in these structures is several times smaller than that in HgTe/HgCdTe and InAs/GaSb/AlSb topological insulators. Therefore, its influence on the quantum spin Hall effect is negligible in strips with a width larger than 150 nm, unless the temperature at which electron transport is measured is less than 1 mK. In the case of quadruple quantum wells, we find the topological insulator phase only when the In content in the interwell barriers is larger than in the external barriers. We show that in these structures, a topological insulator with a bulk energy gap of 0.038 meV can be achieved when the In content in the external barriers is about 0.75. Since this value of the bulk energy gap is very small, quadruple quantum wells are less useful for realizing a measurable quantum spin Hall system, but they are still attractive for achieving a topological phase transition and a nonlocal topological semimetal phase.

## 1. Introduction

The quantum spin Hall effect (QSHE) occurring in 2D topological insulators (TIs) is characterized by gapless helical edge states inside the bulk 2D subband spectrum [1,2]. These edge states are counter-propagating Kramers partners, so backscattering is suppressed and ballistic transport can occur, making 2D TIs interesting candidates for many applications in spintronics and low-power electronics [3,4,5,6,7]. Two-dimensional TIs can also host Majorana bound states when combined with superconductors, which makes them promising materials for topological quantum computing [8,9,10,11,12]. The QSHE was initially predicted for graphene, where the spin–orbit interaction (SOI) produces gaps with opposite signs at the K and K’ points of the hexagonal Brillouin zone [13]. Recently, it has been experimentally confirmed that graphene is a 2D TI with a very small value of the bulk energy gap, $E_{2Dg}^{TI}$, equal to 0.042 meV [14]. A QSHE with much larger values of $E_{2Dg}^{TI}$ than that in graphene was discovered in HgTe/CdTe and InAs/GaSb/AlSb quantum wells (QWs) [15,16,17,18]. In HgTe/CdTe QWs, the QSHE arises from the strong SOI in the HgTe layers, which inverts the subband structure when the QW width exceeds a critical value for the topological phase transition (TPT), i.e., $L_{qw}^{TI}=6$.4 nm [15,16]. For HgTe/CdTe QWs typically grown on CdTe substrates, $E_{2Dg}^{TI}$ reaches about 15 meV and it can be significantly increased up to 55 meV when virtual substrates, introducing compressive stress, are used [19]. In InAs/GaSb two-layer QWs, embedded between AlSb barriers, the TI phase originates from the fact that the valence band (VB) in GaSb is higher than the conduction band (CB) in InAs and the inverted subbands can be obtained by varying the widths of the InAs and GaSb layers [17,18]. In these structures, $E_{2Dg}^{TI}$ is of the order of a few meV and it can be increased to up to 35 meV by adding In to GaSb in InAs/GaInSb QWs [20,21]. A further increase in $E_{2Dg}^{TI}$ to 45 meV was achieved by adding an additional InAs layer to InAs/GaInSb QWs and an $E_{2Dg}^{TI}$ value of 60 meV was theoretically predicted for these three-layer InAs/GaInSb/InAs QWs [22,23].

Two-dimensional TIs have also been proposed theoretically for InN/GaN, InGaN/GaN, and InN/InGaN QWs grown along the [0001] direction [24,25,26]. For these structures, the large built-in electric field originating from the piezoelectric effect and spontaneous polarization may invert the ordering of the CB and VB subbands according to the quantum confined Stark effect (QCSE). This phenomenon, called the polarization-induced TPT, was initially proposed for InN/GaN QWs with $L_{qw}^{TI}$ equal to four atomic monolayers (MLs) [24]. The predicted values of $E_{2Dg}^{TI}$ in these structures depend significantly on the assumed intrinsic SOI in InN, which is still under scientific debate [26,27,28,29]. It was found that $E_{2Dg}^{TI}$ can reach 5 meV when a positive SOI of the order of a few meV is assumed in InN, or it can be about 1.25 meV when a negative SOI in InN is considered [24,25,26]. The disadvantage of InN/GaN QWs is their very large strain of 11%, which dramatically complicates the growth of these structures. In In_x_Ga_1−x_N/GaN and InN/In_y_Ga_1−y_N QWs, strain can be significantly reduced by decreasing the lattice misfit between the materials of barriers and wells. However, decreasing strain causes a reduction in the built-in electric field, which consequently leads to an increase in $L_{qw}^{TI}$ [24,26]. In In_x_Ga_1−x_N/GaN QWs, $L_{qw}^{TI}$ increases faster with decreasing In content in the QWs than the critical thickness for the pseudomorphic growth [24,26]. The situation is more promising in InN/In_y_Ga_1−y_N QWs, where $L_{qw}^{TI}$ increases slower with increasing In content in the barriers, y, than the critical thickness for the pseudomorphic growth [26]. For InN/In_y_Ga_1−y_N QWs with the In content y=0.316 and $L_{qw}^{TI}=1.8\;nm$, $E_{2Dg}^{TI}$ can reach about 2 meV, assuming a negative SOI in InN [26]. Increasing the In content y leads to an increase in $L_{qw}^{TI}$ and a decrease in $E_{2Dg}^{TI}$ [26]. For InN/In_y_Ga_1−y_N QWs with the In content y=0.79 and $L_{qw}^{TI}=4\;nm$, a TI state with $E_{2Dg}^{TI}$ = 0.52 meV was predicted [26].

The TI phase in InGaN-based QWs has not yet been confirmed experimentally due to problems with the growth of topological structures with an inverted order of the CB and VB subbands. The difficulties of growing In_x_Ga_1−x_N/GaN QWs with an In content higher than 0.25 were known almost from the beginning of nitride-based optoelectronics, and for many years they were the main obstacles to obtaining light-emitting diodes and lasers operating in the green and red spectral regions [30,31]. It was realized that the reason for these problems is the existence of biaxial strain in In_x_Ga_1−x_N layers, which hinders the incorporation of In atoms in the In_x_Ga_1−x_N lattice due to the so-called compositional pulling effect [30,31]. Strain was also found to be the main limitation in the growth of ML-thick InN/GaN superlattices [32,33,34,35,36,37,38]. Structures with nominal InN MLs grown on GaN showed much shorter light emission wavelengths, indicating In_x_Ga_1−x_N alloys instead of InN [32,33,34,35]. Measurements performed using high-resolution transmission electron microscopy revealed that ultra-thin nominal InN layers are always some form of In_x_Ga_1−x_N alloy with the In content smaller than 0.35 [35,36,37,38]. Theoretical calculations show a tendency towards ordering in In_x_Ga_1−x_N epitaxial layers caused by local strain effects [39,40,41]. For growth under N-rich conditions, the upper limit of the In content is equal to 0.25 due to repulsive In–In strain interactions, which cause a steep increase in the chemical potential [40,41]. For growth under metal-rich conditions, the chemical potential exhibits a significant nonlinear increase when the In content exceeds 0.33 [41]. Recently, the latter compositional limit has been surpassed in the case of one-ML-thick In_x_Ga_1−x_N/GaN QWs with an In content of approximately 0.45 [42]. A further increase in the In incorporation in In_x_Ga_1−x_N QWs can be achieved by using metamorphic In_x_Ga_1−x_N buffers or InGaN pseudo-substrates, which reduce biaxial compressive strain in these structures [41,43,44,45,46,47]. Much less effort has been devoted to developing the growth of InN/In_y_Ga_1−y_N QWs. In the literature, there are only a few papers that report on the growth of InN/In_y_Ga_1−y_N QWs [48,49,50]. In these structures, the In content y is between 0.8 and 0.9, and the QW widths are equal to 3 and 4.5 nm. It is worth noting that 4.5 nm wide InN/In_0.8_Ga_0.2_N QWs could be close to the TPT, as shown in [26]. However, photoluminescence measurements performed on InN/In_y_Ga_1−y_N QWs showed that the energy gap ($E_{2Dg}$) was about 0.7 eV, suggesting a rather weak contribution from the QCSE [48,49,50]. The weak QCSE in the structures presented in [48,49,50] can originate from screening of the built-in electric field by free carriers created by unintentional doping. Another reason is the use of relatively small barrier widths in [48,49,50], which resulted in a decrease in the built-in electric field in the QWs and an increase in the built-in electric field in the barriers [26].

The possibility of increasing the strength of the QCSE without changing the strain in the system and the thickness of QWs can be achieved in coupled double QWs (DQWs) with thin interwell barriers. This effect was clearly demonstrated by comparing the optical properties of a single In_x_Ga_1−x_N/GaN QW with DQWs having the same compositions and widths of individual wells [51]. It was shown that the recombination energy of interwell indirect excitons in In_0.17_Ga_0.83_N/GaN DQWs with individual QW widths of 2.6 nm is about 350 meV smaller than the recombination energy of excitons in a single QW with the same In content and a thickness of 2.6 nm [51]. This idea of increasing the strength of the QCSE has recently been used in theoretical calculations predicting the TPT in In_x_Ga_1−x_N/GaN and InN/In_y_Ga_1−y_N DQWs [52]. It was shown that the TI phase can be achieved in these structures when the interwell barrier is sufficiently thin and the widths of individual QWs are 2 and 3 MLs, or 3 and 3 MLs [52]. For InN/In_y_Ga_1−y_N DQWs with $L_{qw}^{TI}=3\;MLs$, $E_{2Dg}^{TI}$ can reach about 1.2 meV when the negative SOI in InN is taken into account [52]. However, the In content y that is required to achieve the TI phase in these structures is about 0.33 [52], which is far from the values reached so far in the growth of InN/In_y_Ga_1−y_N QWs [48,49,50].

In this work, we extend the study of the TI phase to coupled InN-based triple and quadruple QWs in which the In content in the interwell barriers, z, is greater than or equal to the In content in the external barriers, y (see Figure 1). We denote these structures as InN/In_z_Ga_1−z_N/In_y_Ga_1−y_N multiple QWs. We consider only structures in which the widths of individual QWs are equal to 2 MLs. We use the eight-band **k∙p** method, assuming that the SOI in InN is negative, which represents the worst-case scenario for achieving the TI phase in these nanostructures. We first investigate triple quantum wells (TQWs) with the same In contents z and y. We show that a TI state with an $E_{2Dg}^{TI}$ of 0.25 meV can appear in TQWs when the widths of interwell barriers ($L_{ib}$) are equal to 2 MLs, whereas for structures with $L_{ib}$ = 3 MLs, the TI phase does not occur. Then, we study TQWs in which the In content z in the interwell barriers is greater than the In content y in the external barriers. We find that increasing the In content z leads to a significant increase in $E_{2Dg}^{TI}$ to about 0.8 meV. In these TQWs, the TI phase can be achieved when the In content y is about 0.64, causing relatively low strain in QWs and making the epitaxial growth of these structures in the range of current technology [42,49]. Next, we generate the effective 2D Hamiltonian and study the QSHE in strip structures containing InN/In_z_Ga_1−z_N/In_y_Ga_1−y_N TQWs. By reducing the width of the strip from 200 to 50 nm, we observe the opening of a gap in the spectrum of the edge states, which is known as the finite size effect and originates from the interaction between the edge states on opposite sides of the strip [53,54,55]. We demonstrate that the finite size effect in InN/In_z_Ga_1−z_N/In_y_Ga_1−y_N TQWs has an oscillatory character regardless of whether the pseudospin-mixing elements of the effective Hamiltonian are omitted or taken into account. We also reveal that the strength of the finite size effect in InN/In_z_Ga_1−z_N/In_y_Ga_1−y_N TQWs is several times smaller than in HgTe/HgCdTe and InAs/GaSb/AlSb QWs; therefore, its influence on the QSHE is negligible in strips wider than 150 nm. In the case of InN/In_z_Ga_1−z_N/In_y_Ga_1−y_N quadruple quantum wells (QQWs), we find that the TI phase appears only when z>y. We show that in these structures, a TI phase with $E_{2Dg}^{TI}=0.038\;meV$ can be reached when the In content y is about 0.75. Since the values of $E_{2Dg}^{TI}$ in QQWs are very small, these structures are less useful for realizing a measurable quantum spin Hall system, but are still attractive for achieving the TPT.

## 2. Theoretical Model

We consider InN/In_z_Ga_1−z_N/In_y_Ga_1−y_N coupled multiple QWs that are grown on virtual substrates having the same In content as external In_y_Ga_1−y_N barriers (see Figure 1). We assume that the thickness of external barriers is large (2000 nm), since this makes the built-in electric field in the QWs extremely large, so the TI phase is easier to achieve [26,52]. For simplicity, we assume that the chemical compositions of interwell barriers are identical. The individual QWs and the interwell barriers are ultra-thin layers, whose widths are expressed in MLs. An ML of InGaN is a double layer of In (Ga) and N atoms, the thickness of which is equal to half of the *c* lattice constant [24]. Therefore, the width of a QW ($L_{qw}$) and the width of an interwell barrier ($L_{ib}$) can be expressed by the following formula:(1)$$L_{l}=\frac{1}{2}n_{l}c_{l}\left(1-R_{B,l}\varepsilon_{xx,l}\right),\;l=qw,\;ib$$
where $n_{l}$ is the number of MLs in the layer, $c_{l}$ denotes the *c*-lattice constant of unstrained material of the layer, $R_{B,l}$ is the biaxial relaxation coefficient, $\varepsilon_{xx,l}=\frac{a_{s}}{a_{l}}-1$ is the in-plane strain, and $a_{l}$ and $a_{s}$ are the *a*-lattice constants of unstrained materials of the layer and the substrate, respectively. We took the lattice constants for InN and GaN from [56] and assumed that for InGaN alloys, they depend linearly on composition [57]. The $R_{B}$ coefficient is given as follows:(2)$$R_{B}=\frac{1}{\varepsilon_{xx}}\left[1-\sqrt{1+\frac{2}{C_{333}}\left(-c+\sqrt{c^{2}-2C_{333}d}\right)}\right]$$
where $c=C_{33}+2C_{133}\left(\varepsilon_{xx}+\frac{1}{2}\varepsilon_{xx}^{2}\right)$ and $d=2C_{13}\left(\varepsilon_{xx}+\frac{1}{2}\varepsilon_{xx}^{2}\right)+\left(C_{113}+C_{123}\right){\left(\varepsilon_{xx}+\frac{1}{2}\varepsilon_{xx}^{2}\right)}^{2}$ [58,59,60]. In the above formula, $C_{13}$ and $C_{33}$ are the second-order elastic constants, while $C_{113}$, $C_{123}$, $C_{133}$, and $C_{333}$ are the third-order elastic constants [60]. We took the elastic constants for GaN and InN from [60], while for InGaN alloys, we assumed linear dependences on composition for the third-order elastic constants and took the nonlinear composition dependences of the second-order elastic constants from [61,62]. The built-in electric field in the multiple QW structure is calculated taking into account the first- and second-order piezoelectric effects and the spontaneous polarization [63,64]. We assume that the structures are undoped, so screening of the built-in electric field by free carriers is not considered.

In order to calculate the electronic states in InN/In_z_Ga_1−z_N/In_y_Ga_1−y_N multiple QWs, we apply the **k·p** method with the 8-band Hamiltonian $H^{8\times 8}$, which includes relativistic and nonrelativistic linear-wave-vector terms [26,28]. This Hamiltonian takes into account the negative SOI in InN, which was predicted via ab initio calculations performed using the quasiparticle self-consistent GW method [26,28]. For the bulk 3D crystals, the Hamiltonian $H^{8\times 8}$ has the following form:(3)$$H^{8\times 8}=\left[\begin{array}{cccccccc} H_{c} & -Q & Q^{*} & R & 0 & 0 & 0 & 0\\ -Q^{*} & F & K^{*} & M_{-}^{*} & 0 & 0 & -W^{*} & 0\\ Q & K & G & -N_{+} & 0 & -W^{*} & -T & \sqrt{2}{∆}_{3}\\ R & M_{-} & -N_{+}^{*} & L & 0 & 0 & \sqrt{2}{∆}_{3} & -S^{*}\\ 0 & 0 & 0 & 0 & H_{c} & Q^{*} & -Q & R\\ 0 & 0 & -W & 0 & Q & F & K & -M_{+}\\ 0 & -W & -T^{*} & \sqrt{2}{∆}_{3} & -Q^{*} & K^{*} & G & N_{-}^{*}\\ 0 & 0 & \sqrt{2}{∆}_{3} & -S & R & -M_{+}^{*} & N_{-} & L \end{array}\right]\begin{array}{c} \left.|iS,↑\right\rangle \\ \left.|-\left(X+iY\right)/\sqrt{2},↑\right\rangle \\ \left.|\left(X-iY\right)/\sqrt{2},↑\right\rangle \\ \left.|Z,↑\right\rangle \\ \left.|iS,↓\right\rangle \\ \left.|\left(X-iY\right)/\sqrt{2},↓\right\rangle \\ \left.|-\left(X+iY\right)/\sqrt{2},↓\right\rangle \\ \left.|Z,↓\right\rangle \end{array}$$
where $H_{c}=E_{vb}+E_{g}+A_{c⟘}k_{⟘}^{2}+A_{c||}k_{z}^{2}$, $Q=P_{2}k_{+}/\sqrt{2}$, $R=P_{1}k_{z}$, $F={∆}_{1}+{∆}_{2}+\left(A_{2}+A_{4}\right)k_{⟘}^{2}+\left(A_{1}+A_{3}\right)k_{z}^{2}$, $G=F-2{∆}_{2}$, $L=A_{2}k_{⟘}^{2}+A_{1}k_{z}^{2}$, $K=A_{5}k_{+}^{2}$, $M_{+}=\left[A_{6}k_{z}+i\left(A_{7}+\alpha_{4}\right)\right]k_{+}$, $M_{-}=\left[A_{6}k_{z}-i\left(A_{7}+\alpha_{4}\right)\right]k_{+}$, $N_{+}=\left[A_{6}k_{z}+i\left(A_{7}-\alpha_{4}\right)\right]k_{+}$, $N_{-}=\left[A_{6}k_{z}-i\left(A_{7}-\alpha_{4}\right)\right]k_{+}$, $S=i\alpha_{1}k_{+}$, $T=i\alpha_{2}k_{+}$, and $W=i\left(\alpha_{1}+\alpha_{3}\right)k_{+}$. The top valence band energy and the energy gap are denoted by $E_{vb}$ and $E_{g}$, respectively. $A_{c⟘}$ and $A_{c||}$ describe the dispersion of the CB, whereas $P_{1}$ and $P_{2}$ are the Kane parameters [25,26]. The valence band parameters $A_{1},\ldots ,A_{7}$, $\alpha_{1},\ldots ,\alpha_{4}$, and ${∆}_{1},\ldots ,{∆}_{3}$ were taken from [28] assuming linear dependences on composition in InGaN alloys. Additionally, the parameters $A_{1},\ldots ,A_{6}$ were rescaled according to [25]. Strain and the built-in electric field were included in the Hamiltonian $H^{8\times 8}$ according to [65,66]. Since for QWs grown along the [0001] crystallographic direction, $k_{z}$ is not a good quantum number, we replaced it with the operator $-i\frac{\partial }{\partial z}$. The standard symmetrization of operators containing the product of functions and derivatives was used to ensure the Hermiticity of $H^{8\times 8}\left({\vec{k}}_{⟘},k_{z}=-i\frac{\partial }{\partial z}\right)$ [25]. The subband dispersion in InN/In_z_Ga_1−z_N/In_y_Ga_1−y_N multiple QWs is obtained by numerically solving the eigenvalue problem with the Hamiltonian $H^{8\times 8}\left({\vec{k}}_{⟘},k_{z}=-i\frac{\partial }{\partial z}\right)$ [26,67].

To check the applicability of the 8-band **k∙p** method to determine the TPT in 2ML-thick InGaN-based coupled multiple QWs, we performed calculations of the energy gap ($E_{g}$) in short-period In_x_Ga_1−x_N/GaN superlattices (SLs) and compared the results to those obtained via ab initio calculations [68,69] and photoluminescence measurements [70]. We chose SLs with the widths of QWs and barriers equal to 2 MLs and we assumed that the structures were grown pseudomorphically on GaN substrates. Periodic boundary conditions for the wavefunctions were used in the calculations of the $E_{g}$ in SLs [71]. In Figure 2, we compare the $E_{g}$ obtained using the Hamiltonian $H^{8\times 8}$ (a solid line) with the ab initio results (squares) and experimental data (dot) taken from [68,69,70]. One can see that the **k∙p** method determines the $E_{g}$ well for In_0.33_Ga_0.67_N/GaN SLs. For larger In contents, the **k∙p** method slightly overestimates the values of $E_{g}$ compared to the ab initio calculations. In general, however, a satisfactory agreement between these two approaches is reached.

In order to calculate the electronic states in a strip of finite width, we apply the effective 2D Hamiltonian $H_{eff}$, which describes the subband dispersion near the energy gap [25,26]. The basis of the Hamiltonian $H_{eff}$ consists of six eigenstates corresponding to three doubly degenerate levels of the Hamiltonian $H^{8\times 8}\left({\vec{k}}_{⟘}=0,\;k_{z}=-i\frac{\partial }{\partial z}\right)$, i.e., the lowest CB level and the highest light-hole (LH) and heavy-hole (HH) levels. These states have well-defined projections of the total angular momentum onto the *z* axis equal to $\pm \frac{1}{2}$ for the CB and LH states and $\pm \frac{3}{2}$ for the HH states. The Hamiltonian $H_{eff}$ has the following form:(4)$$H_{eff}=\begin{array}{c} \begin{array}{cccccc} \left.|CB,1/2\right\rangle & \left.|LH,-1/2\right\rangle & \left.|HH,3/2\right\rangle & \left.|CB,-1/2\right\rangle & \left.|LH,1/2\right\rangle & \left.|HH,-3/2\right\rangle \end{array}\\ \left[\begin{array}{cccccc} E_{0}+E_{1}k_{⟘}^{2} & C_{2}k_{-} & C_{1}k_{+} & iC_{3}k_{-} & iMk_{⟘}^{2} & iB_{2}k_{-}^{2}\\ C_{2}k_{+} & L_{0}+L_{1}k_{⟘}^{2} & B_{1}k_{+}^{2} & -iMk_{⟘}^{2} & iC_{4}k_{+} & iC_{5}k_{-}\\ C_{1}k_{-} & B_{1}k_{-}^{2} & H_{0}+H_{1}k_{⟘}^{2} & {-iB}_{2}k_{-}^{2} & iC_{5}k_{-} & 0\\ -iC_{3}k_{+} & iMk_{⟘}^{2} & {iB}_{2}k_{+}^{2} & E_{0}+E_{1}k_{⟘}^{2} & {-C}_{2}k_{+} & -C_{1}k_{-}\\ -iMk_{⟘}^{2} & -iC_{4}k_{-} & -iC_{5}k_{+} & {-C}_{2}k_{-} & L_{0}+L_{1}k_{⟘}^{2} & B_{1}k_{-}^{2}\\ {-iB}_{2}k_{+}^{2} & -iC_{5}k_{+} & 0 & {-C}_{1}k_{+} & B_{1}k_{+}^{2} & H_{0}+H_{1}k_{⟘}^{2} \end{array}\right] \end{array}$$
where $E_{0}$, $L_{0}$, and $H_{0}$ are the energies of the states $\left.|CB,\pm 1/2\right\rangle$, $\left.|LH,\pm 1/2\right\rangle$, and $\left.|HH,\pm 1/2\right\rangle$, respectively. The coefficients $B_{1}$, $C_{1}$, and $C_{2}$ determine the coupling between states with the same pseudospins, whereas the coefficients $B_{2}$, $C_{3}$, $C_{4}$, $C_{5}$, and M describe the coupling between states with the opposite pseudospins [25]. The coupling between states with different pseudospins is due to the structural asymmetry of the QW potential, which originates from the built-in electric field (see Figure 1) [25]. All of the coefficients of the Hamiltonian $H_{eff}$ are obtained by applying the mini-band **k·p** method and the Löwdin perturbation approach [25,26]. We assume that the strip is oriented along the *x* axis, so $k_{y}$ is not a good quantum number and we replace it with the operator $-i\frac{\partial }{\partial y}$. The dispersion of electronic states in a strip is obtained by numerically solving the eigenvalue problem with the Hamiltonian $H_{eff}\left(k_{x},\;k_{y}=-i\frac{\partial }{\partial y}\right)$ [26,67].

## 3. Results and Discussion

As we mentioned in the introduction, we investigated the QSHE effect in InN/In_z_Ga_1−z_N/In_y_Ga_1−y_N TQWs and QQWs in which the In content in the internal barriers, z, was greater than or equal to the In content in the external barriers, y. The widths of individual QWs were equal to 2 MLs. We first studied InN/In_z_Ga_1−z_N/In_y_Ga_1−y_N TQWs with z=y. We considered two types of structures with $L_{ib}$ equal to 2 MLs (2–2–2–2–2) and 3 MLs (2–3–2–3–2). In Figure 3, we present $E_{2Dg}$ as a function of the In content y and the subband dispersions for distinct phases occurring in these structures. For the structures with $L_{ib}=2\;MLs$, we observe that a decrease in the In content y causes TPT from the normal insulator (NI) to the TI phase via the Weyl semimetal (WSM) phase. For the NI phase, the CB subband is above the LH subband; for the WSM phase, both subbands touch at finite ${\vec{k}}_{⟘}$; for the TI phase, the LH subband is above the CB subband. TPT occurs in the structure with the In content $y_{TPT}=0.30675$. In the TI phase, $E_{2Dg}^{TI}$ reaches a maximum value of $E_{2Dg,max}^{TI}=0.255\;meV$ at the In content $y_{TI,max}=0.3058$. For this structure, the amplitude of the built-in electric field in QWs $\left|F_{qw}\right|$ is equal to 10.87 MV/cm. For an In content y smaller than $y_{NTSM}=0.3036$, we obtain a nonlocal topological semimetal (NTSM), arising from nonlocal overlapping between the LH and CB subbands [25,26]. For structures with $L_{ib}=3MLs$, the TI phase does not occur. In these structures, we find that reducing the In content y below 0.3029 causes a non-topological phase transition from the NI phase to the nonlocal nontopological semimetal (NNSM), characterized by normal ordering of the CB and LH subbands [52]. Further decreasing y results in a TPT from the NNSM to the NTSM via the buried Weyl semimetal (BWSM) phase [52]. The phase transitions described above are similar to the phase transitions found in InN/In_y_Ga_1−y_N DQWs [52]. In particular, for very similar In contents in the barriers, the TI phase appeared in InN/In_y_Ga_1−y_N DQWs with $L_{qw}=3\;MLs$ [52]. The obvious advantage of InN/In_z_Ga_1−z_N/In_y_Ga_1−y_N TQWs over InN/In_y_Ga_1−y_N DQWs is that in the former case, the TI occurs at $L_{qw}=2\;MLs$. However, a significant disadvantage of InN/In_y_Ga_1−y_N/In_y_Ga_1−y_N TQWs is a much smaller $E_{2Dg,max}^{TI}$ compared to InN/In_y_Ga_1−y_N DQWs, for which a value of about 1.2 meV was reached [52].

To increase $E_{2Dg}^{TI}$ in TQWs, we propose to use InN/In_z_Ga_1−z_N/In_y_Ga_1−y_N TQWs in which z>y. We focus on structures with $L_{ib}=2\;MLs$. In Figure 4, we show $E_{2Dg}$ as a function of the In content y for structures with an In content z equal to 0.5, 0.6, 0.7, 0.8, and 0.9. In all of these cases, we find the TI phase between the WSM and the NTSM. For the InN/In_0.5_Ga_0.5_N/In_y_Ga_1−y_N TQWs (Figure 4a), $E_{2Dg,max}^{TI}$ is equal to 0.523 meV, which is twice as large as the $E_{2Dg,max}^{TI}$ obtained for InN/In_z_Ga_1−z_N/In_y_Ga_1−y_N TQWs with z=y. $E_{2Dg,max}^{TI}$ increases with increasing the In content z from 0.5 to 0.8, reaching the largest value equal to 0.832 meV for InN/In_0.8_Ga_0.2_N/In_y_Ga_1−y_N TQWs (see Figure 4d). For InN/In_0.9_Ga_0.1_N/In_y_Ga_1−y_N TQWs, we find that $E_{2Dg,max}^{TI}$ is equal to 0.796 meV. The obtained values of $E_{2Dg,max}^{TI}$ are large enough to allow experimental verification of the QSHE in TQWs [14,72]. The values of $y_{TPT}$ and $y_{TI,max}$ increase monotonically with increasing z. We note that for InN/In_0.9_Ga_0.1_N/In_y_Ga_1−y_N TQWs, $y_{TI,max}$ reaches 0.6345, causing a relatively low strain in the QW layers of about 0.0367 and making the epitaxial growth of these structures within the range of current technology [42,49]. The properties of the TI phase in InN/In_z_Ga_1−z_N/In_y_Ga_1−y_N TQWs are presented overall in Figure 5. Figure 5a shows that $y_{TPT}$ and the amplitude of strain in the QWs at the TPT, $\left|\varepsilon_{xx,qw}^{TPT}\right|$, depend almost linearly on z. In Figure 5b, we demonstrate the values of $E_{2Dg,max}^{TI}$ and the window of the In content for the TI phase, ${∆y}_{TI}=y_{TPT}-y_{NTSM}$, as a function of z. We can see that both of these quantities depend non-monotonically on z. For smaller values of z, the increases in $E_{2Dg,max}^{TI}$ and ${∆y}_{TI}$ are related to the increase in the difference between the LH and HH energy levels at ${\vec{k}}_{⟘}=0\;$(denoted by ${∆E}_{LH-HH}$) and to the decrease in the amplitude of the built-in electric field in QWs $\left|F_{qw}\right|$, as shown in Figure 5c. We note that the decrease in $\left|F_{qw}\right|$ increases the overlap of the wavefunctions of the CB and LH states due to the QCSE, reducing the tendency for nonlocal overlapping between the CB and LH subbands. For z=0.9, the reductions in $E_{2Dg,max}^{TI}$ and ${∆y}_{TI}$ are associated with smaller quantum confinement and leakage of the CB wavefunction into the external barrier, which decreases the overlap between the wavefunctions of the CB and LH states and increases the nonlocal overlapping between the CB and LH subbands.

In order to study the QSHE occurring in the TI phase, we applied the effective 2D Hamiltonian $H_{eff}$ and computed the electronic states in strip structures containing InN/In_z_Ga_1−z_N/In_y_Ga_1−y_N TQWs. We chose the InN/In_0.8_Ga_0.2_N/In_0.5684_Ga_0.4316_N TQWs for which we obtained the largest value of $E_{2Dg,max}^{TI}$. To determine the coefficients of the Hamiltonian $H_{eff}$, we used 150 states of the Hamiltonian $H^{8\times 8}\left({\vec{k}}_{⟘}=0,\;k_{z}=-i\frac{\partial }{\partial z}\right)$; namely, 25 doubly degenerate states for the CB, LH, and HH subbands. We obtained the following values of the coefficients of the Hamiltonian $H_{eff}$: $E_{0}=0.745696$ eV, $L_{0}=0.748218$ eV, $H_{0}=0.743508$ eV, $E_{1}=3.525123\;eV\cdot {Å}^{2}$, $L_{1}=-18.907530\;eV\cdot {Å}^{2}$, $H_{1}=-18.825562\;eV\cdot {Å}^{2}$, $B_{1}=16.652953\;eV\cdot {Å}^{2}$, $C_{1}=-0.189983\;eV\cdot Å$, $C_{2}=0.187564\;eV\cdot Å$, $B_{2}=-0.849472\;eV\cdot {Å}^{2}$, $C_{3}=0.017855\;eV\cdot Å$, $C_{4}=-0.068348\;eV\cdot Å$, $C_{5}=-0.003012\;eV\cdot Å$, and $M=1.152510\;eV\cdot {Å}^{2}$. The amplitudes of the linear ($C_{3}$, $C_{4}$, $C_{5}$) and quadratic ($B_{2}$, M) coefficients describing the coupling between states with the opposite pseudospins are significantly smaller than the amplitudes of the linear ($C_{1}$, $C_{2}$) and quadratic ($B_{1}$) coefficients determining the coupling between states with the same pseudospins. The coefficients $C_{3}$, $C_{4}$, $C_{5}$, $B_{2}$, M were neglected in several papers on topological InN/GaN QWs [24,73,74], although they were found to play a key role in determining the TPT in these structures [25]. Below, we will show that they are important for determining the spectrum of the edge states in strip structures.

In Figure 6a, we compare the subband dispersions in InN/In_0.8_Ga_0.2_N/In_0.5684_Ga_0.4316_N TQWs obtained using the **k·p** Hamiltonian $H^{8\times 8}$ (red lines) and the Hamiltonian $H_{eff}$ (black lines). We can see that the Hamiltonian $H_{eff}$ describes quite well the dispersions of the LH, CB, and HH subbands in the small vicinity of ${\vec{k}}_{⟘}=0$, which is similar to the results obtained for HgTe/CdTe TQWs [75]. However, the Hamiltonian $H_{eff}$ significantly overestimates $E_{2Dg}^{TI}$. Therefore, it is not suitable for studying the phase transitions in InN/In_z_Ga_1−z_N/In_y_Ga_1−y_N TQWs. Despite this disadvantage, it can be used to investigate edge states with small wavevectors, occurring in strips of finite thickness. In Figure 6b,c, we show the dispersion of electronic states in strips with widths $L_{strip}$ of 200 nm and 50 nm. For a strip with $L_{strip}=200\;nm$, one can see that the Dirac point of the edge state dispersion curve is located in the bulk band gap. For a strip with $L_{strip}=50\;nm$, we observe a gap in the spectrum of the edge states ($E_{edg}$). The appearance of $E_{edg}$ originates from the interaction between the edge states on opposite sides of the strip and is called the finite size effect [53,54,55]. Figure 6d demonstrates $E_{edg}$ on a logarithmic scale as a function of $L_{strip}$. The solid symbols (in red) correspond to the results obtained using the full Hamiltonian $H_{eff}$, while the open symbols (in black) represent the results obtained by neglecting the pseudospin-mixing elements of the Hamiltonian $H_{eff}$ (i.e., assuming $B_{2}=C_{3}=C_{4}=C_{5}=M=0$). One can see that the finite size effect in InN/In_z_Ga_1−z_N/In_y_Ga_1−y_N TQWs has an oscillatory character regardless of whether the pseudospin-mixing elements of the effective Hamiltonian are omitted or taken into account. A similar situation was found for InAs/GaSb/AlSb QWs, for which, as in the case of InN-based QWs, there is rather small overlapping between the wavefunctions of the CB and VB subbands [55]. The period of oscillations of $E_{edg}$ and their rate of decay depend significantly on the inclusion of the pseudospin-mixing elements of the Hamiltonian $H_{eff}$. We found that for strips with $L_{strip}\geq 150\;nm$, $E_{edg}$ is smaller than $8.62\times 10^{-5}\;meV,$ which corresponds to a characteristic temperature $T^{*}=E_{edg}/k_{B}$ that is smaller than 1 mK. Therefore, we state that when $L_{strip}\geq 150\;nm$, the finite size effect is negligible and has no influence on the current flow along the edges unless the temperature at which the experiment is performed is less than 1 mK. In Figure 6e, we compare $E_{edg}$ and $E_{2Dg}$ on a linear scale as a function of $L_{strip}$. Note that a tenfold magnification of the values of $E_{edg}$ has been used. One can see that $E_{edg}$ is at least twenty times smaller than $E_{2Dg}$. For HgTe/HgCdTe and InAs/GaSb/AlSb QWs, $E_{edg}$ is at least five and four times smaller than $E_{2Dg}$ when $L_{strip}\geq 50\;nm$ [55]. Additionally, for HgTe/HgCdTe QWs, the condition that $E_{edg}$ is less than $8.62\times 10^{-5}\;meV$ (i.e., $T^{*}\leq 1\;mK$) is fulfilled when $L_{strip}\geq 600\;nm$ [54]. Therefore, we conclude that the strength of the finite size effect in InN/In_z_Ga_1−z_N/In_y_Ga_1−y_N TQWs is several times smaller than in HgTe/HgCdTe and InAs/GaSb/AlSb QWs.

Finally, we extended our study to InN/In_z_Ga_1−z_N/In_y_Ga_1−y_N QQWs. We considered only structures with the widths of individual QWs and interwell barriers equal to 2MLs. For QQWs with z=y, we found that the TI phase does not occur and the phase transitions are similar to the case of TQWs with $L_{ib}=3\;MLs$, as shown in Figure 3a,c. For QQWs with z>y, the TI phase can appear between the WSM and the NTSM, as for TQWs with $L_{ib}=2\;MLs$. In Figure 7a, $E_{2Dg}$ for InN/In_0.9_Ga_0.1_N/In_y_Ga_1−y_N QQWs is shown as a function of the In content y. We can see that for these structures, the TI occurs when y is about 0.753, which is very close to InN/In_0.8_Ga_0.2_N QWs grown using plasma-assisted molecular-beam epitaxy [49]. Unfortunately, we found that $E_{2Dg,max}^{TI}=0.038\;meV$, which is more than 20 times smaller than the value of $E_{2Dg,max}^{TI}$ obtained for InN/In_0.9_Ga_0.1_N/In_y_Ga_1−y_N TQWs (see Figure 4e) and is also smaller than $E_{2Dg}^{TI}$ in graphene [14]. Figure 7b shows that $y_{TPT}$ and $\left|\varepsilon_{xx,qw}^{TPT}\right|$ change linearly with increasing the In content z in InN/In_z_Ga_1−z_N/In_y_Ga_1−y_N QQWs. Figure 7c demonstrates that $E_{2Dg,max}^{TI}$ and ${∆y}_{TI}$ increase linearly with increasing z, which originates from the increase in ${∆E}_{LH-HH}$ and the decrease in $\left|F_{qw}\right|$ in these structures, as shown in Figure 7d. For InN/In_z_Ga_1−z_N/In_y_Ga_1−y_N QQWs, $E_{2Dg,max}^{TI}$ and ${∆y}_{TI}$ are very small; therefore, we do not consider these nanostructures as candidates for an experimental verification of the QSHE and the TI phase. However, InN/In_z_Ga_1−z_N/In_y_Ga_1−y_N QQWs can be very useful for achieving the TPT and the NTSM phase due to large values of $y_{TPT}$ and $y_{NTSM}$, in comparison to topological TQWs.

## 4. Conclusions

We have presented a theoretical proposal to obtain the TI phase and the QSHE in coupled InN/In_z_Ga_1−z_N/In_y_Ga_1−y_N TQWs with the widths of individual QWs and interwell barriers equal to 2 MLs. We have shown that for structures with the same In contents z and y, a TI state with an $E_{2Dg,max}^{TI}$ of 0.25 meV can appear. Increasing the In content z in TQWs with z>y leads to a significant increase in $E_{2Dg,max}^{TI}$ to about 0.8 meV. The TI phase in these structures can be achieved when the In content y is about 0.64, which results in relatively low strain in QWs and makes the epitaxial growth within the range of current technology [42,49]. We have demonstrated that the finite size effect in these structures has an oscillatory character regardless of whether the pseudospin-mixing elements of the effective Hamiltonian are omitted or taken into account. The strength of the finite size effect in InN/In_z_Ga_1−z_N/In_y_Ga_1−y_N TQWs is several times smaller than in HgTe/HgCdTe and InAs/GaSb/AlSb QWs; therefore, its influence on the QSHE is negligible in strips wider than 150 nm. In the case of InN/In_z_Ga_1−z_N/In_y_Ga_1−y_N QQWs, we found that the TI phase appears only when z>y. In these structures, the TI state with $E_{2Dg}^{TI}=0.038\;meV$ can be reached when the In content y is about 0.75. Since the values of $E_{2Dg}^{TI}$ in QQWs are very small, these structures are less useful for realizing a measurable quantum spin Hall system, but are still attractive for achieving the TPT and the NTSM. We hope that the presented results will guide future experimental studies to the discovery of the topological phases in group-III nitride nanostructures and contribute to new applications of these prospective topological nanomaterials.

## Figures and Tables

**Figure 1 nanomaterials-13-02212-f001:**
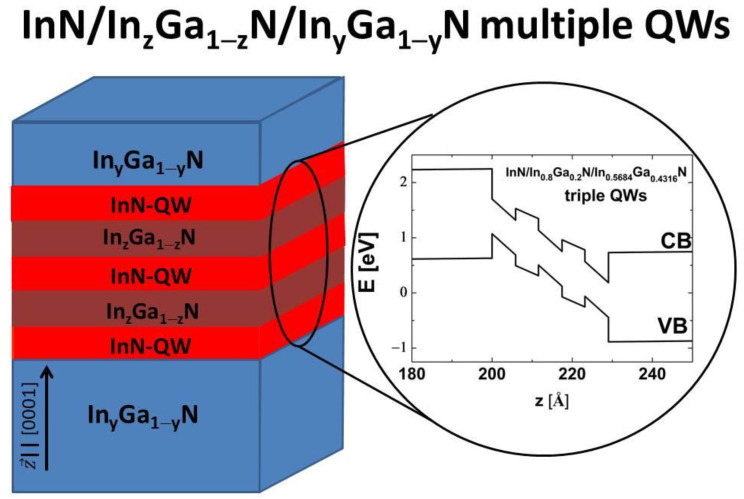
Schematic representation of InN/In_z_Ga_1−z_N/In_y_Ga_1−y_N coupled multiple QWs. The right-hand side shows the CB and VB edge profiles for an exemplary structure containing InN/In_0.8_Ga_0.2_N/In_0.5684_Ga_0.4316_N triple QWs with the widths of individual QWs and interwell barriers equal to 2 MLs.

**Figure 2 nanomaterials-13-02212-f002:**
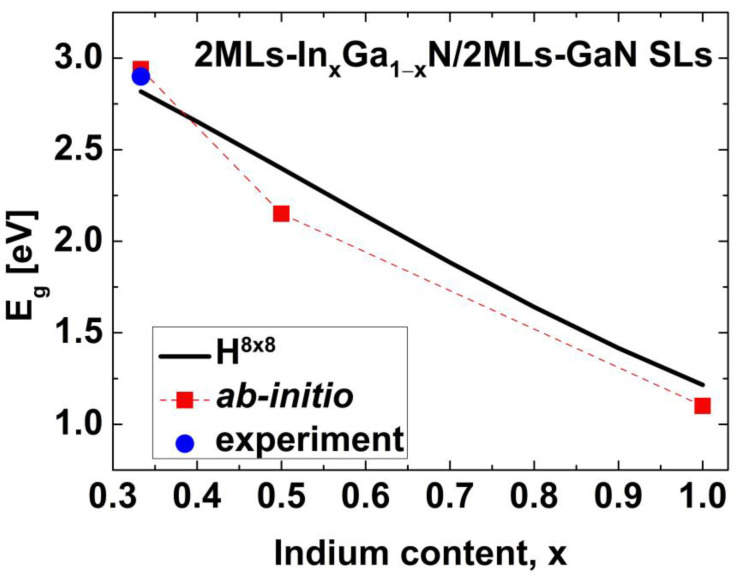
*E_g_* for 2 MLs-In_x_Ga_1−x_N/2MLs-GaN SLs as a function of the In content x. The solid line corresponds to the results obtained using the 8-band **k∙p** method, while squares and the dot represent the results of ab initio calculations and photoluminescence measurements taken from [68,69,70].

**Figure 3 nanomaterials-13-02212-f003:**
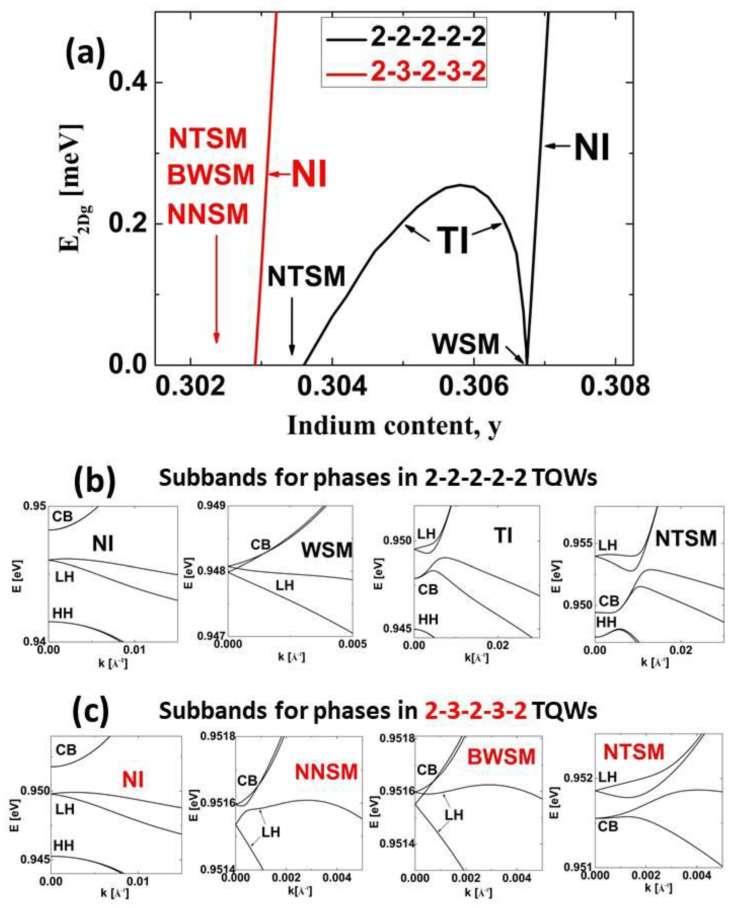
(**a**) *E*_2*Dg*_ for InN/In_z_Ga_1−z_N/In_y_Ga_1−y_N TQWs with z=y as a function of the In content y. The widths of individual QWs are equal to 2MLs and the widths of interwell barriers are equal to 2 MLs (2–2–2–2–2) and 3 MLs (2–3–2–3–2). (**b**,**c**) Subband dispersions for the phases in 2–2–2–2–2 TQWs (**b**) and in 2–3–2–3–2 TQWs (**c**). The phases appear in order of decreasing In content in the barriers.

**Figure 4 nanomaterials-13-02212-f004:**
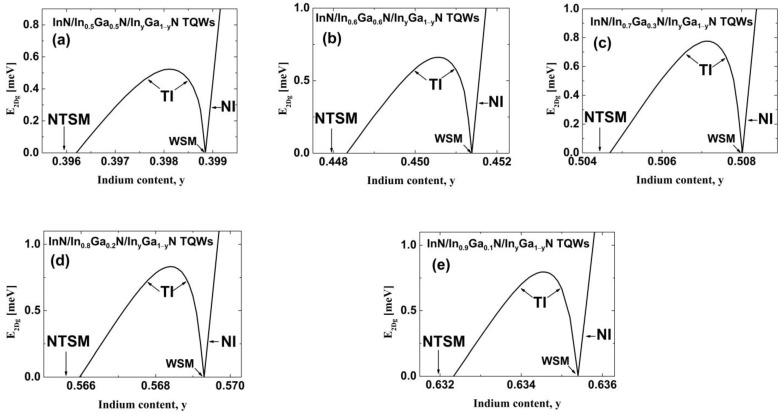
$E_{2Dg}$ for InN/In_z_Ga_1−z_N/In_y_Ga_1−y_N TQWs with different In content z in the interwell barriers equal to 0.5 (**a**), 0.6 (**b**), 0.7 (**c**), 0.8 (**d**), and 0.9 (**e**), as a function of the In content y in the external barriers. The widths of individual QWs and interwell barriers are equal to 2 MLs.

**Figure 5 nanomaterials-13-02212-f005:**
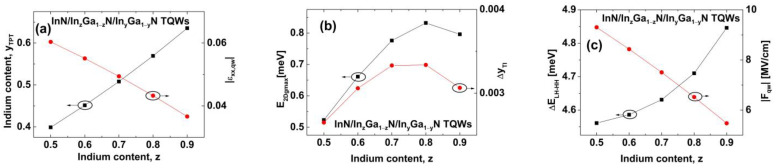
(**a**) Values of $y_{TPT}$ (squares) and $\left|\varepsilon_{xx,qw}^{TPT}\right|$ (dots) for InN/In_z_Ga_1−z_N/In_y_Ga_1−y_N TQWs as a function of the In content z. (**b**) Values of $E_{2Dg,max}^{TI}$ (squares) and ${∆y}_{TI}$ (dots) for InN/In_z_Ga_1−z_N/In_y_Ga_1−y_N TQWs as a function of the In content z. (**c**) Values of ${∆E}_{LH-HH}$ (squares) and $\left|F_{qw}\right|$ (dots) for InN/In_z_Ga_1−z_N/In_y_Ga_1−y_N TQWs as a function of the In content z.

**Figure 6 nanomaterials-13-02212-f006:**
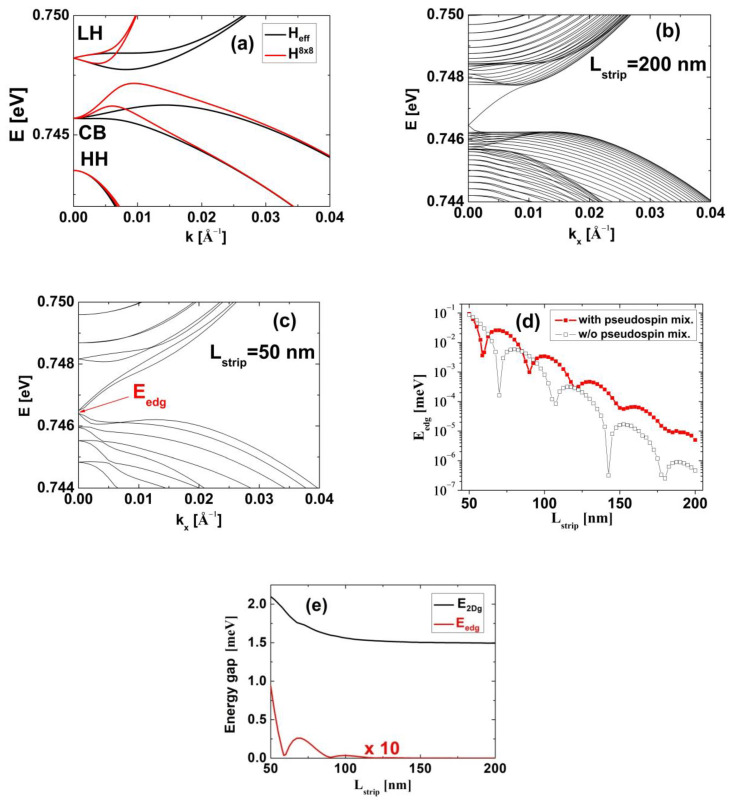
(**a**) Subband dispersions in topological InN/In_0.8_Ga_0.2_N/In_0.5684_Ga_0.4316_N TQWs, obtained using the 8-band **k·p** Hamiltonian $H^{8\times 8}$ (red lines) and the effective 2D Hamiltonian $H_{eff}$ (black lines). (**b**,**c**) Dispersion of electronic states in strips containing InN/In_0.8_Ga_0.2_N/In_0.5684_Ga_0.4316_N TQWs and having widths $L_{strip}$ equal to 200 nm (**b**) and 50 nm (**c**). (**d**) $E_{edg}$ (on a logarithmic scale) obtained with and w/o the pseudospin mixing elements of the Hamiltonian $H_{eff}$ as a function of $L_{strip}$. (**e**) $E_{edg}$ and $E_{2Dg}$ (on a linear scale) as a function of $L_{strip}$. Note that the values of $E_{edg}$ are magnified ten times.

**Figure 7 nanomaterials-13-02212-f007:**
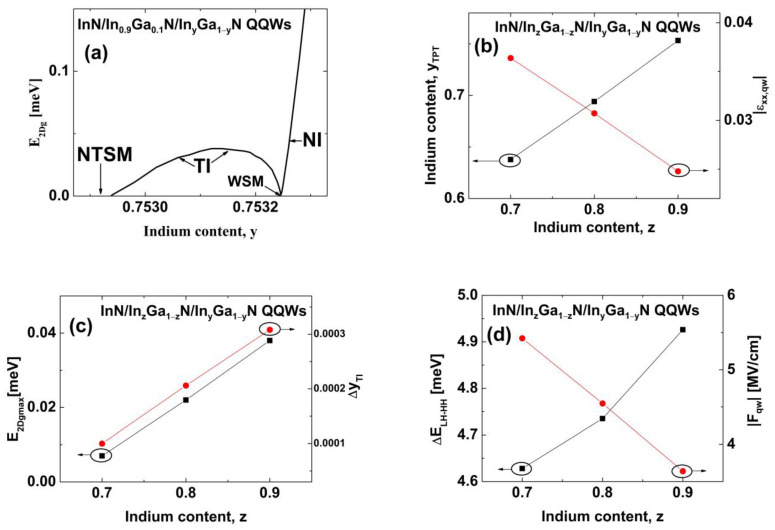
(**a**) $E_{2Dg}$ for InN/In_0.9_Ga_0.1_N/In_y_Ga_1−y_N QQWs as a function of the In content y. The widths of individual QWs and interwell barriers are equal to 2 MLs. (**b**) Values of $y_{TPT}$ (squares) and $\left|\varepsilon_{xx,qw}^{TPT}\right|$ (dots) for InN/In_z_Ga_1−z_N/In_y_Ga_1−y_N QQWs as a function of the In content z. (**c**) Values of $E_{2Dg,max}^{TI}$ (squares) and ${∆y}_{TI}$ (dots) for InN/In_z_Ga_1−z_N/In_y_Ga_1−y_N QQWs as a function of the In content z. (**d**) Values of ${∆E}_{LH-HH}$ (squares) and $\left|F_{qw}\right|$ (dots) for InN/In_z_Ga_1−z_N/In_y_Ga_1−y_N QQWs as a function of the In content z.

## Data Availability

The data underlying this article are available from the corresponding author on reasonable request.

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
