# Peer review of "Quantum Spin Hall Effect in Two-Monolayer-Thick InN/InGaN Coupled Multiple Quantum Wells"

_nanomaterials, 2023, doi:10.3390/nano13152212_

Round 1
Reviewer 1 Report (Previous Reviewer 3)
No comments
No comments
Author Response
I thank the referee for the positive evaluation of my paper. I submitted the manuscript to the MDPI editing service. The linguistic corrections were taken into account in the revised manuscript.
Reviewer 2 Report (Previous Reviewer 1)
The manuscript can be published now.
The manuscript can be published now.
Author Response
I thank the referee for the positive evaluation of my paper. I submitted the manuscript to the MDPI editing service. The linguistic corrections were taken into account in the revised manuscript.
This manuscript is a resubmission of an earlier submission. The following is a list of the peer review reports and author responses from that submission.
Round 1
Reviewer 1 Report
(1) In the abstract, the authors determined that in bands with widths greater than 150 nm, the effect on the quantum spin Hall effect is negligible by the results that the intensity of the effective size effect in the structure is several times smaller than the intensity in the GaSb topological limb, and it is recommended to briefly explain it here.
(2) Please note the typography of the format, and the formula and picture 3 should be in the middle of the stylistic part.
(3)It is suggested that the discussion of the built-in electric field variables be added to the discussion of the results, which will make the conclusions more logical.
(4) In the results and discussions, it is concluded that the strength of the effective size effect in InN etc. is many times smaller than the intensity of the finite size effect in the GaSb quantum potential well?
(5) After carefully reading the full text, I did not find the author's definition of "ML", can you explain it in detail?
very good
Reviewer 2 Report
The manuscript describes a theoretical study of the quantum spin Hall effect in InN/InGaN coupled multiple quantum wells with the individual well widths equal to two atomic monolayers. Authors consider triple and quadruple quantum wells in which the In content in the interwell barriers is greater than or equal to the In content in the external barriers. Since the TI phase in InGaN-based QWs has not been confirmed experimentally, the presented results might be important for future experimental studies to discover the topological phases in group-III nitride nanostructures.
The model seems to be correct. The numerical calculations can not be verified by Referee, however, physical results sound reasonable.
Author Response
I would like to thank the referee for the very positive evaluation of my paper.
Reviewer 3 Report
The Introduction section is too extensive, usually a summary of the essence of the article within 100-200 words is required.
The submitted contribution is theoretical, while no assessment is given of the advantage or disadvantage of a small value of the effective band gap for realisation of TI conditions, so that one can be guided by this assessment for the design of new semiconductor structures and devices based on them.
I pay the attention again that the submitted contribution is theoretical, however in the Conclusion section the author focuses on experimentally implemented structures.
"Introduction" should be corrected in terms of improving the presentation in English.
